# Development of the Chromatographic Method for Simultaneous Determination of Azaperone and Azaperol in Animal Kidneys and Livers

**DOI:** 10.3390/ijms24010100

**Published:** 2022-12-21

**Authors:** Izabella Kośka, Paweł Kubalczyk

**Affiliations:** 1Department of Environmental Chemistry, Faculty of Chemistry, University of Lodz, Pomorska 163, 90-236 Lodz, Poland; 2Doctoral School of Exact and Natural Sciences, University of Lodz, Banacha 12/16, 90-237 Lodz, Poland

**Keywords:** azaperone, azaperol, sedatives, high-performance liquid chromatography, animal kidneys, animal livers

## Abstract

A precise and accurate method for the simultaneous determination of azaperone and azaperol in meat tissues has been developed. This paper describes the first method to be so fast, simple, and useful, especially for many laboratories that do not have sophisticated equipment. This method is based on LC separation and UV-Vis detection. During the sample preparation, the meat tissue was homogenized in acetonitrile at a ratio of 1:4 (tissue weight:acetonitrile volume). The homogenate was centrifuged, the supernatant was evaporated in a lyophilizator, and then the evaporation residue was dissolved in 20 µL of ethanol. For deproteinization, 15 µL of perchloric acid was added, and the sample prepared in this way was injected into a chromatographic column and analyzed using reversed-phased HPLC. The mobile phase consisted of 0.05 mol/L phosphate buffer pH 3.00 (component A) and acetonitrile (component B). UV detection was conducted at 245 nm. The experimentally determined LOQs were 0.25 µg/kg for azaperone and 0.12 µg/kg for azaperol. For both analytes, the calibration curves showed linearity in the tested concentration range from 50 to 300 µg/kg of tissue. The accuracy of the presented method did not exceed 15%, and the recovery was in the range of 85–115%. A validated analytical procedure was implemented for the analysis of various animal tissues for their content of azaperone and azaperol.

## 1. Introduction

Among the drugs administered to animals, sedative drugs can be distinguished. They are administered to animals, mainly pigs, during transport to the slaughterhouse [1,2,3,4]. Pigs are extremely susceptible to stress, and breeders use these drugs to minimize the risk of animal death and to maintain a high quality of meat, because meat from stressed animals is very hard and tasteless [5]. Due to the widespread use of these compounds, drug residues are sometimes present in food. The reason for this is often nonadherence to the withdrawal period, when the administration of veterinary drugs has not been stopped for an appropriate amount time before the slaughter of the animal or the collection of, for example, eggs. Eating meat with residues of chemicals is dangerous to the health of the consumers [1,6,7]. The most commonly used sedative drug is azaperone (AZN) (Figure 1A), or 1-(4-fluorophenyl)-4-[4-(2-pyridin-yl)-1-piperazinyl]-1-butanone, belonging to the butyrophenones group [3].

In animals, AZN is reduced and metabolized to azaperol (AZL) [8] (Figure 1B).

As many as 10% of these compounds are believed to have neuroleptic strength, and the remaining 90% are subject to a maximum residue limit (MRL).

The MRL for AZN and AZL residues is 100 µg/kg for pig liver and kidneys according to the European Union Commission Regulation (EC) No. 1958/98 as an amendment to the Council Regulation (EEC) No. 2377/90 [8,9,10]. To identify meat from dishonest producers, it is very important to control the level of the residues of these drugs in the meat bought and eaten by consumers, and to check that the content of these compounds does not exceed the MRL level. For this reason, it is necessary to develop fast, cheap, sensitive, precise, and, above all, simple analytical methods that will enable the determination of AZN and AZL at low concentrations, using simple analytical tools that most laboratories are equipped with. This paper describes a method for the simultaneous determination of AZN and AZL in animal tissues. In the literature, methods are described that allow the determination of AZN in tissues by LC-MS-MS [1,5,6,11,12,13] and immunochromatographic assay (ICA) [14], in animal urine [6] and in tissues [8,15] with the use of LC-FLD, in biological fertilizer blood meal with the use of LLE-LC-MS [16], and in blood meal with the use of ASE and LC–Orbitrap MS [17]. In the method presented here, we decided to use the HPLC technique with UV-Vis detection because we want to develop a useful and universal method. We focused on simplifying the sample preparation procedure as much as possible, to allow this procedure to be reproduced by other scientists or units which would like to test meat quality, even in less sophisticated laboratories. Taking the above into account, the method presented stands out among other protocols described in the literature primarily because of its great usability, but also because of its simplicity while maintaining satisfactory validation parameters. This methodology is also not time consuming, which is an additional advantage of method here described.

## 2. Results and Discussion

### 2.1. Sample Preparation

#### 2.1.1. Optimization of the Ratio of Animal Tissue Mass to Acetonitrile Volume

During the optimization of homogenization step, different tissue mass to acetonitrile volume ratios were checked. The following ratios were selected for optimization: 1:4, 1:5, 1:10, 1:20, and 1:40. It can be seen in the Figure 2 that the highest and the most reproducible signals were obtained when the kidney tissue sample was homogenized with acetonitrile in a ratio of 1:4.

It can be assumed that a further increase in kidney mass in relation to the volume of acetonitrile would increase the signals even more; however, increasing the amount of meat prevented homogenization, because the sample was too dense. Therefore, a 1:4 ratio of tissue mass to acetonitrile volume was selected.

#### 2.1.2. Selection of the Lyophilization Temperature

The next step was to choose the temperature of the samples lyophilization. The stability of AZN and AZL was checked at the following temperatures: 40 °C, 50 °C, 60 °C, 70 °C, and 80 °C. The results depicted in the Figure 3 show that the analytes are stable at 40 °C and 50 °C, but that, above the temperature of 50 °C, the signals obtained from the analytes decrease. Another advantage of lyophilizing samples at 50 °C is the fact that the evaporation time is shortened (~20 min), and therefore the time required to complete the sample preparation is also shorter, which is extremely advantageous. Moreover, the results were better in this case, i.e., there were higher analytical signals and the reproducibility was satisfactory. 

#### 2.1.3. Optimization of Ethanol Volume after Lyophilization

In order to select the appropriate volume of ethanol to dissolve the residue after evaporation, the size of the analytical signal was checked in relation to the volume of the ethanol. For this purpose, the sample after evaporation was dissolved in 10 µL, 20 µL, 30 µL, 40 µL, and 50 µL of ethanol. It was important to minimize this volume as much as possible to avoid excessive dilution of the sample, and, at the same time, to maintain a good reproducibility of the analytical signals. The highest signals were obtained for 10 µL of ethanol; unfortunately, this volume is too small to repeatably mix the sample in a 2 mL Eppendorf tube. Hence, a small quantity of the evaporation residue can remain on the inner wall of the tube. When residue was dissolved in 20 µL of ethanol, only a small decrease in the height of the analytical signal was observed (Figure 4); furthermore this greatly facilitates mixing. Finally, we decided to choose 20 µL of the ethanol to efficiently dissolve the evaporation residue.

#### 2.1.4. Optimization of Perchloric Acid Volume

PCA, trichloroacetic acid, or excess acetonitrile can be used to deproteinize the samples. Since PCA is one of the most popular deproteinization agents, we decided to use it in our methodology. This is a very important step in the sample preparation, because the proteins present in a sample could easily block the chromatography column. At the same time, it is essential not to add an unnecessary excess of deproteinizing reagent, as it further dilutes the sample, resulting in a poorer LOQ. For this purpose, we checked the volume of PCA at which complete deproteinization occurs. The following volumes of 3 mol/L PCA were tested: 5, 10, 15, 20, 25, and 30 µL. After the addition of 15 µL of PCA, we noticed that adding another volume of PCA did not cause further protein precipitation. Therefore, this volume of acid was selected for the experiments to avoid excessive dilution of the sample and ensure its complete deproteinization.

### 2.2. Chromatographic Conditions

One of the most used analytical techniques for the analysis of biological samples is reversed-phase liquid chromatography. Since the selection of the chromatographic conditions is crucial to the quality of the analyte separation, several parameters have been studied, including the type of mobile phase, pH, flow rate, gradient profile, and temperature.

#### 2.2.1. Selection of Mobile Phase

Various compositions of mobile phases were tested. As component A was tested: phosphate buffer, formic acid, and acetic acid in different pH. As component B was tested, acetonitrile was used in each case. It was observed that, for a phosphate buffer, the peaks are higher than otherwise. Therefore, we decided to perform a pH dependence of 0.05 mol/L phosphate buffer to check the effect of the pH of the mobile phase on the retention of the analytes. It was decided to use such a concentration of phosphate buffer because it is a concentration high enough to keep the pH of the mobile phase constant but also not high enough to crystallize in the column. The following pH values were verified in the buffering range of 0.05 mol/L phosphate buffer: 3, 6, 7, 8, 11, 12. The highest peaks and, at the same time, the most reproducible results of the analyses were obtained when a pH of 3 for the mobile phase was used. Therefore, 0.05 mol/L phosphate buffer with pH 3.00 (component A) and acetonitrile (component B) was chosen to serve as the mobile phase.

During the development of the method, both isocratic and gradient elution were considered. The chromatographic separation of AZN and AZL was achieved using gradient elution. When the isocratic elution was used, the peaks of the analytes did not separate from each other, and these peaks did not separate satisfactorily from other components of the sample. The best chromatographic separation of AZN and AZL was achieved using gradient elution. Several of the gradient profiles checked during development are presented in Table 1.

As can be seen in Figure 5, the use of the gradient profile 0–7 min, 10–50% B; 7–9 min, 50–10% B (gradient number 10) allows the highest analytical signals for both AZN and AZL. After a gradient run, the column was equilibrated with the starting concentration of the mobile phase for 1 min prior to the loading of next sample.

#### 2.2.2. Optimization of Separation Temperature

The influence of the column temperature on the height of the analytical signals was checked. For this purpose, the column oven was heated to the following temperatures: 25 °C, 30 °C, 35 °C, and 40 °C. We realized that the changes in the column temperature did not significantly affect the height of the analytes’ peaks or improve the separation, but merely led to narrower peaks and minimally shortened the analysis time. Therefore, all of the analyses were performed at room temperature.

### 2.3. Calibration and Other Validation Data

Validation parameters such as LOD, LOQ, the precision of the method, and its accuracy were determined in accordance with the FDA criteria for the analysis of biological samples [18]. The LOD and LOQ were determined experimentally using the signal-to-noise method. The concentration of an analyte that is equal to the LOD gives a signal three times higher than the baseline noise. The LOQ is the concentration of an analyte for which the signal height corresponds to nine times the height of the baseline noise. The LOD values evaluated for AZN and AZL were 1.0 µg/kg tissue and 0.4 µg/kg tissue, respectively. The LOQ values were 2.5 µg/kg for AZN and 1.2 µg/kg for AZL. Both the LOD and LOQ values for the method are lower than the MRL, which will allow the determination of the analytes at concentration levels similar to the MRL. The LOD and LOQ are similar to those in the LC-MS/MS method [1], and lower than those in the LC-MS/MS [7], LC-FLD [8], and LC-UV [8] methods. Detailed data are presented in Table 2.

Calibration curves for AZN and AZL in meat tissues were constructed for five concentrations in the range from 50, 75, 100, 200, and 300 µg/kg of tissue, and each series was performed in triplicate. The concentrations were selected in such a way that the calibration curve would allow us to estimate whether a given meat sample exceeds the MRL. The calibration curves obtained by the method described showed linearity in the whole concentration range studied. The square of the linear correlation coefficient (R^2^) for AZN was 0.9985, and that for AZL R^2^ was 0.9991. The equation of the calibration curve for AZN was y = (0.0222 ± 0.0005)x + (0.2904 ± 0.0843), while for AZL it was y = (0.0510 ± 0.0009)x + (1.1217 ± 0.1484).

The coefficient of variation (CV) of the points on the calibration curve for AZN was in the range from 0.8 to 10.6%, and that for AZL was from 0.7 to 5.3%. The recovery was in the range from 97.1% to 107.3% for AZN, and from 96.4 to 105.1% for AZL. The values that describe the calibration curves are consistent with the FDA criteria required for the analysis of biological samples [18]. The next step in the research was to check the intra-day and inter-day precision and accuracy of the method. For this purpose, three concentrations were selected (the first concentration represented the beginning of the calibration curve, the middle concentration was taken from the middle of the calibration curve, and the third concentration was near the end of the calibration curve). Meat tissue samples were prepared at the concentrations indicated above. The method precision for AZN and AZL does not exceed 15%, while the method accuracy is in the range of 85–115%. These values are at satisfactory levels; moreover, taking into account the simplicity of the sample preparation step, we believe that the method we developed can be used to determine the content of AZN and AZL in routine analyses of meat tissues. Any laboratory with basic equipment would be able to perform such analyses. All of the validation data are presented in Table 3.

An unquestionable advantage of the described analytical procedure is the simplicity of the sample preparation. During the development of the sample preparation procedure, the focus was on its usability. The procedure for the sample preparation was developed in such a way that this method could be used in any analytical laboratory where meat is tested; therefore, the method is simple and also not very time consuming. The total sample preparation time is 45 min; however, it should be taken into account that, during this time, as many samples can be prepared as the number of available places in the centrifuge and lyophilizator rotor allows (in the case of our laboratory where the procedure was developed, this is 48 samples at one time). Analyses of meat tissues for their AZN and AZL content using the method described require less time than those using the previously described LC–MS/MS [19], SPE-LC–MS/MS [2,20], and LLE-LC–MS/MS methods [21].

Despite the many advantages of this method, it has disadvantages, too. Its weaknesses include, above all, the use of large volumes of the toxic solvent acetonitrile, as well as a small concentration sensitivity when using UV-Vis detection that requires concentrating the sample. Table 4 contains the basic parameters describing a similar method taken from the literature [22] and the method presented here. However, the data in the Table 4 show that, despite having fewer steps in its preparation of samples, our methodology yielded more satisfactory LOD results.

### 2.4. Application to Real Samples

The validated procedure described in this paper was used to determine AZN and AZL in meat tissues, i.e., pork kidneys and livers. Samples 1–6 are kidneys, and samples 7–10 are livers. All samples were purchased from local meat breeders. The tissue samples were spiked with a known amount of AZN and AZL to give a concentration of 230 µg/kg tissue for both analytes and prepared as described in the “Sample collection and preparation” section. The samples were then analyzed using HPLC. All data presented the results of the assays are summarized in Table 5. The results obtained and collected in Table 3 indicate that the described methodology can be successfully used for routine analysis of meat for its content of AZP and AZL. The results thus obtained are repeatable and consistent. Representative chromatograms obtained for the tissue sample and the spiked tissue sample are shown in Figure 6. In developing this method, we used a large number of different animal tissue samples that had been purchased over a long period of time, and we did not notice any interference. Azaperone is the sedative drug most commonly used in animals for this purpose. Unfortunately, we do not have standards for other drugs in this group to check their potential interference. However, it is very unlikely that an animal would receive several sedatives at the same time. In addition to sedatives, farm animals may also receive other medications, including antibiotics, such as fluoroquinolones, to treat infections or prevent them. Several of these drugs (i.e., ofloxacin and ciprofloxacin) were checked by us for potential chromatogram interference, and we did not notice any. However, if the tissue samples contain a large number of interfering agents, a modification of the extraction step could be necessary during sample preparation. The CV values do not exceed 15%. This method can be successfully applied to the analysis of real samples for AZN and AZL content.

## 3. Materials and Methods

### 3.1. Instruments

The Agilent 1220 Infinity HPLC system (Agilent Technologies, Waldbronn, Germany) coupled with the diode-array detector and equipped with a binary pump, degasser, automatic injector, and column oven was used to perform all of the experiments. Separation was performed on the Zorbax SB C-18 chromatographic column (150 × 4.6 mm, 5 µm, Agilent Technologies, Waldbronn, Germany). The peaks corresponding to the analytes were assigned by comparing both the diode-array spectra and the retention times recorded for the real samples with the matching set of data achieved for authentic compounds. For instrument control, data acquisition, and quantitative analysis, the OpenLAB ChemStation Edition software was used. The Millipore Milli-Q-RG System (Waterford, Ireland) deionizer was used for water purification. Deionized water (Type 1) was obtained with a resistivity of 18 kΩ·cm at 25 °C. The water was filtered using a membrane filter with a pore diameter of 0.22 μm. The pH meter (Mettler-Toledo, Greifensee, Switzerland) was used to adjust the pH of the buffer solutions, for proteins removal a centrifuge with a fast cooling function (Mikro 220R, Hettich Zentrifugen, Tuttlingen, Germany) was applied, and the Labconco CentriVap (Kansas, MO, USA) was used to lyophilize the samples.

### 3.2. Chemicals

The standards of the analytes, i.e., azaperone (C_19_H_22_FN_3_O) and azaperol (C_19_H_24_FN_3_O) were from Sigma Aldrich (Saint Louis, MO, USA). Sodium phosphate dibasic (Na_2_HPO_4_), sodium dihydrogen phosphate (NaH₂PO₄), trisodium phosphate (Na_3_PO_4_), and acetonitrile were purchased from Sigma (Steinheim, Germany). Perchloric acid (HClO_4_) and ethanol (99.8%) were obtained from POCH (Gliwice, Poland).

### 3.3. Chromatographic Conditions

For the chromatographic separation of AZN and AZL, a reversed-phase Zorbax SB C-18 (150 × 4.6 mm, 5 µm) column was applied. The mobile phase consisted of 0.05 mol/L phosphate buffer pH 3 (component A) and acetonitrile (component B). The chromatographic separation of AZN and AZL from the other components of the sample was achieved using gradient elution: 0–7 min 10–50% B; 7–9 min 50–10% B. All analyses were performed with a constant flow rate of the mobile phase of 1 mL/min and at room temperature. UV-Vis detection at 245 nm for both analytes was used. The peaks were identified through the comparison of the retention times and the diode-array spectra, taken at the real time of the analysis, with the corresponding set of data obtained by analyzing authentic compounds.

### 3.4. Sample Collection and Preparation

Animal tissues, such as kidneys, were purchased at markets and local stores. To prepare each sample, 0.5 g of tissue was placed in a 3 mL polypropylene tube with 2 mL of acetonitrile and homogenized. The homogenate was centrifuged at 13,680× *g* (12,000 rpm) for 15 min, and then the supernatant was collected and transferred to a 2 mL polypropylene tube. The sample was lyophilized at 50 °C, which led to the evaporation of acetonitrile, and then the residue was dissolved in 20 µL of ethanol. For deproteinization, 15 µL of PCA was added, the sample was centrifuged at 13,680× *g* (12,000 rpm) for 10 min, and then 25 µL of supernatant solution was collected in chromatographic vial. Finally, 5 µL of the sample prepared in this way was injected into the column and analyzed using HPLC.

### 3.5. Method Validation

After the optimization of all parameters, the method was validated. The validation parameters, such as the limit of detection (LOD), the limit of quantification (LOQ), and the precision and accuracy of the method, were determined according to the Food and Drug Administration (FDA) criteria for analytical procedures and method validation [18].

### 3.6. Calibration of the Method

Stock solutions of 1 mg/mL AZN and AZL were prepared by dissolving appropriate amounts of the compounds in 1 mL of ethanol. To perform the calibration, standard solutions were prepared in three series in the concentration range of 50–300 µg/kg tissue. Calibration solutions were prepared according to the following procedure: 0.5 g of kidney tissue was placed in a polypropylene tube with 2 mL of acetonitrile, and then the sample was homogenized and spiked with an ethanolic solution of the analytes in an appropriate concentration. The homogenate was centrifuged at 13,680× *g* (12,000 rpm) for 15 min, and, next, the supernatant was transferred to a 2 mL polypropylene tube. Each sample was lyophilized at 50 °C, and then the evaporation residue was dissolved in 20 µL of ethanol. Subsequently, 15 µL of PCA was added for deproteinization, the sample was centrifuged and 25 µL of supernatant solution was collected, and 5 µL of the sample was injected into a chromatographic column and analyzed. After the analysis, the peak heights of AZN and AZL were plotted against the corresponding concentrations of analytes and the calibration curves were fitted using a least-squares linear regression analysis.

### 3.7. Greenness

Green analytical chemistry focuses on making analytical procedures more environmentally benign and safer to humans [23]. We decided to check how the proposed method looks in this regard. We used very smart software dedicated for this purpose, i.e., the Analytical GREEnness calculator for the assessment of greenness of analytical procedures based on the SIGNIFICANCE principles [23]. The calculated greenness of the presented method is 0.58 (Figure 7). Our procedure is based on combined sample homogenization and analytes extraction and on the separation of analytes by HPLC with UV detection. The procedure consisted of an external sample treatment with a reduced number of steps (principle 1), and 0.5 g of tissue sample is needed (principle 2). The measurement is off-line (principle 3), and the procedure involves four distinct steps, such as homogenization, lyophilization, centrifugation, and separation (principle 4). The procedure is semi-automated and miniaturized (principle 5). During the analysis, no derivatization step was required (principle 6). The analytical wastes include 2 mL of acetonitrile for homogenization, 20 µL of ethanol for dilution, 15 µL of PCA for deproteinization, and 10.1 mL (including 2.9 mL of acetonitrile) of the HPLC mobile phase (principle 7). Two analytes are determined in a single run, and the sample throughput is ~4 samples per hour, if we assume that about 48 samples can be prepared simultaneously (principle 8). The lyophilization system is the most energy-demanding analytical technique in our protocol (principle 9). Some of the reagents can be obtained from bio-based sources (principle 10). The procedure requires 4.9 mL of toxic solvents (principle 11), and acetonitrile is considered explosive (fumes) and highly flammable (principle 12).

An important question to ask is whether our methodology is better or worse in terms of greenness compared to other methods for the determination of azaperone and azaperol in animal tissues. Unfortunately, in the works compared, the authors do not specify the degree of greenness of their procedures. Therefore, we tried to estimate the greenness of these methods using a dedicated calculator and the data available in the articles. As can be seen on Figure 8, our methodology (score 0.58) is comparable in terms of greenness to the HPLC-FL (score 0.53) method [8] and better than the HPLC-MS/MS (score 0.40) [5] and HPLC-UV (score 0.48) [22] methods.

Our procedure achieved a better score, mainly because it does not use a complicated and multi-step liquid-liquid extraction or SPE. Efforts should continue to be made to improve procedures in order to offset their negative impacts on the environment and human health.

## 4. Conclusions

A simple and cheap chromatographic procedure has been developed to simultaneously determine AZN and AZL in meat tissues. The preparation of the tissue samples is very quick and involves sample homogenization and deproteinization, as well as concentration of the analytes. The procedure is simple and does not require sophisticated equipment; it is based on HPLC separation with UV-Vis detection. Both the sample preparation, and the chromatographic analysis cause this method to stand out among other HPLC methods for determining these analytes in animal tissues. Apart from its simplicity and speed of execution, the method presented is characterized by high sensitivity and precision. Due to the very high precision and accuracy of the method, we strongly believe that it can be used in the future for the routine analysis of meat for its content of azaparone, which is very often administered to animals, and its metabolite azaperol. The described method can be used both in veterinary medicine and in food safety testing. Since eating the residues of these compounds with meat when the withdrawal period has not been respected is hazardous to the health of consumers, the method may be helpful in protecting human health. 

## Figures and Tables

**Figure 1 ijms-24-00100-f001:**
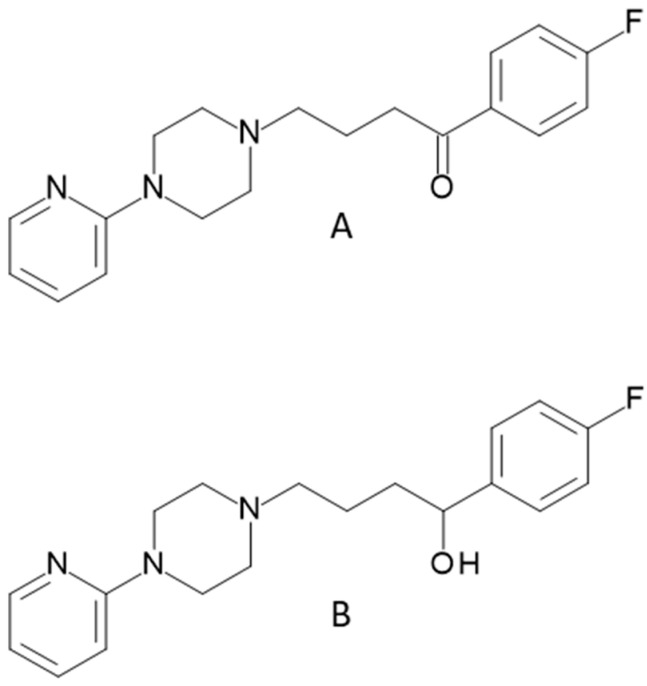
Structure of azaperone (**A**) and azaperol (**B**).

**Figure 2 ijms-24-00100-f002:**
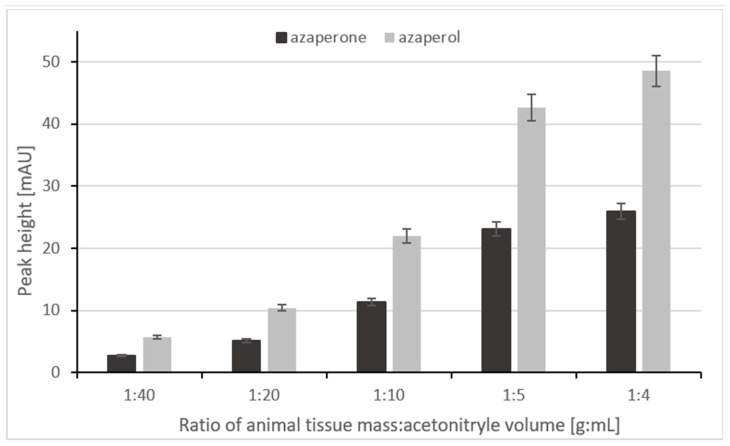
The relationship between peak height and the ratio of animal tissue mass to acetonitrile volume.

**Figure 3 ijms-24-00100-f003:**
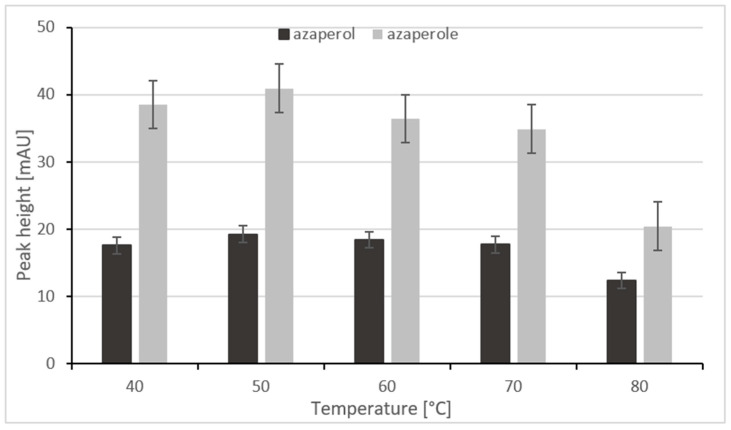
The relationship between the peak height and the temperature during lyophilization.

**Figure 4 ijms-24-00100-f004:**
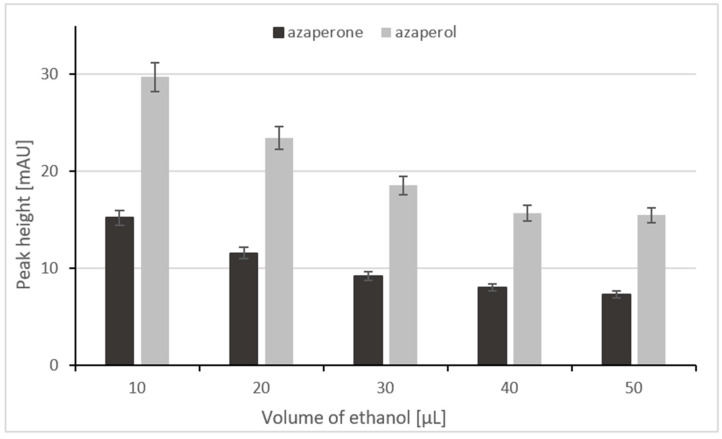
The relationship between the peak height and the volume of ethanol.

**Figure 5 ijms-24-00100-f005:**
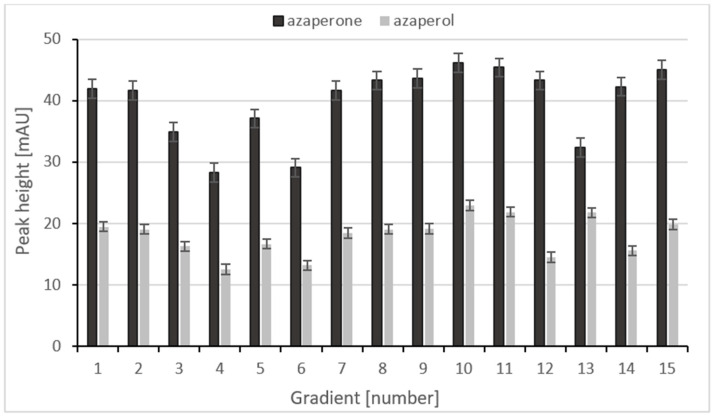
The relationship between the peak height and the gradient.

**Figure 6 ijms-24-00100-f006:**
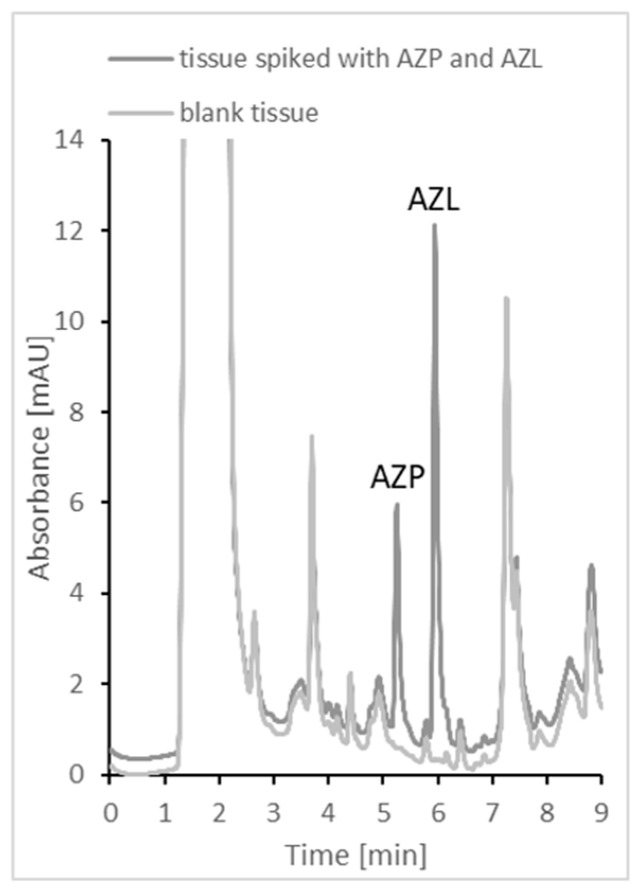
Representative chromatograms of blank tissue and tissue spiked with AZP and AZL (250 µg/kg tissue). Chromatographic conditions are as described in Section 3.3.

**Figure 7 ijms-24-00100-f007:**
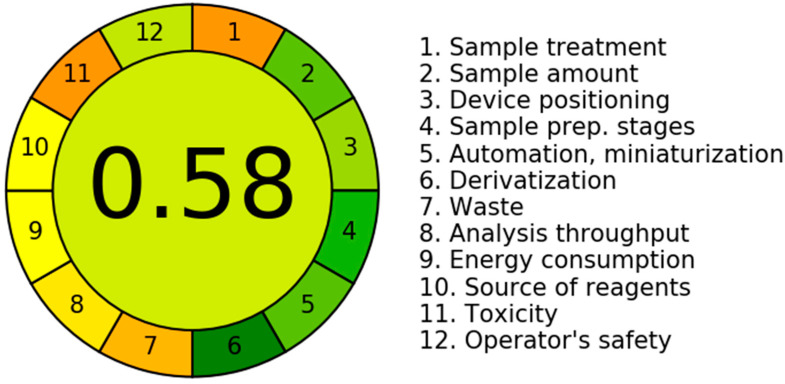
Calculated greenness of the method.

**Figure 8 ijms-24-00100-f008:**
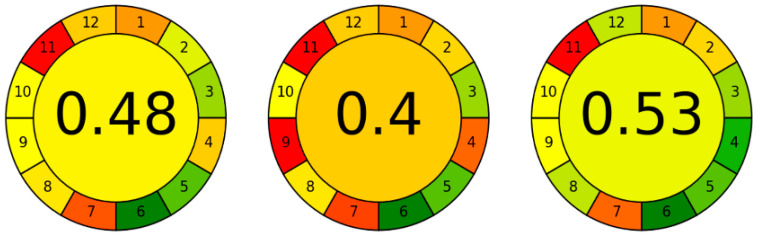
Comparison of greennesses of the methods, from the left [5,8,22].

**Table 1 ijms-24-00100-t001:** Gradient profiles checked during development method.

Number of Gradient	Gradient Profile
1	0–0.5 min, 15% B; 0.5–5 min, 15–50% B; 5–9 min, 50–15% B
2	0–0.5 min, 15% B; 0.5–4 min, 15–50% B; 4–9 min, 50–15% B
3	0–0.5 min, 15% B; 0.5–5 min, 15–50% B; 5–6 min, 50% B; 6–9 min, 50–15% B
4	0–0.5 min, 15% B; 0.5–5 min, 15–30% B; 5–9 min, 30–15% B
5	0–0.5 min, 15% B; 0.5–5 min, 15–30% B; 5–7 min, 30–50% B; 7–10 min, 50–15% B
6	0–0.5 min, 15% B; 0.5–5 min, 15–30% B; 5–6 min, 30% B; 6–10 min, 30–15% B
7	0–0.5 min, 15% B; 0.5–5 min, 15–40% B; 5–10 min, 40–15% B
8	0–0.5 min, 15% B; 0.5–5 min, 15–40% B; 5–7 min, 40–50% B; 7–10 min, 50–15% B
9	0–0.5 min, 15% B; 0.5–5 min, 15–40% B; 5–6 min, 40% B; 6–10 min, 40–15% B
10	0–7 min, 10–50% B; 7–9 min, 50–10% B
11	0–1 min, 10–15% B; 1–5 min, 10–15% B; 5–9 min, 50–10% B
12	0–0.5 min, 20% B; 0.5–5 min, 20–50% B; 5–9 min, 50–20% B
13	0–1 min, 20–30% B; 1–5 min, 30–50% B; 5–9 min, 50–20% B
14	0–1 min, 15–20% B; 1–5 min, 20–50% B; 5–9 min, 50–15% B
15	0–1 min, 20–30%; 1–5 min, 30–50% B; 5–9 min, 50–12% B

**Table 2 ijms-24-00100-t002:** Comparison the LOD, LOQ, R^2^, precision and accuracy values of the described method with published methods.

Method	LOD (µg/kg)		LOQ (µg/kg)		R^2^	Precision [%]	Accuracy [%]
	AZN	AZL	AZN	AZL			
LC-MS/MS [2]	0.06–0.1	0.1	0.2–0.4	0.4	>0.99	<15	74.2–91.8
LC-MS/MS [7]	0.5	0.5	2.5	2.5	0.9826–0.9965	1.1–16.6	69.8–85.5
LC-FLD [8]	10	3	10	5	>0.99	<11.0	88.2–91.2
LC-UV [15]	1	no data	no data	no data	0.997	0.6–14.6	97.0–112.9
Presented method	1.0	0.4	2.5	1.2	0.9985–0.9991	2.6–9.9	93.4–109.5

**Table 3 ijms-24-00100-t003:** Validation data.

Added * [µg/kg]	Intra-Day			Inter-Day		
	Found ± SD [µg/kg]	CV [%]	Accuracy [%]	Found ± SD [µg/kg]	CV [%]	Accuracy [%]
Azaperone						
80	78.51 ± 2.60	3.3	98.1	81.51 ± 4.50	5.5	101.9
150	156.59 ± 5.20	3.3	104.4	158.09 ± 4.50	2.9	105.4
250	273.71 ± 18.20	6.7	109.5	257.19 ± 20.64	8.0	102.9
Azaperol						
80	74.74 ± 4.93	6.6	93.4	75.39 ± 6.30	8.4	94.2
150	159.05 ± 9.06	5.7	106.0	151.21 ± 11.32	7.5	100.8
250	272.12 ± 7.07	2.6	108.9	257.74 ± 25.59	9.9	103.1

* n = 3.

**Table 4 ijms-24-00100-t004:** Comparison of the presented method with a similar method described in the literature.

	Method Described in the Literature [22]	Presented Method
Mobile phase	acetonitrile—0.025% aqueous diethylamine mixture (2:3, *v*/*v*)	0.05 mol/L phosphate buffer pH 3 (component A) and acetonitrile (component B)
Column/Stationary phase	ODS column (Asahipak ODP-50 4D, 150 mm × 4.6 mm, Showa Denko K.K., Kanagawa, Japan)	Zorbax SB C-18 (150 mm × 4.6 mm, 5 µm, Agilent Technologies).
Linearity	0.05–2 µg/mL	50–300 µg/kg of tissue
Applications	Tissues analysis	Tissues analysis
Merits	the method is applicable, good accuracy and precision, confirmation with LC/MS	not time consuming, easy to perform, does not require sophisticated equipment, good accuracy, good precision, lower LOD
Demerits	more steps in sample preparation, higher LOD, large volumes of toxic acetonitryle, UV-Vis detection (small concentration sensitivity)	large volumes of toxic acetonitryle, UV-Vis detection (small concentration sensitivity)

**Table 5 ijms-24-00100-t005:** Determination of AZN and AZL in tissues (livers, kidneys)—results.

Sample Number	Added * [µg/kg]	Found ± SD [µg/kg]	CV [%]
Azaperone			
1	230.00	218.15 ± 9.38	4.3
2	230.00	222.65 ± 2.60	1.2
3	230.00	225.66 ± 4.50	2.0
4	230.00	214.40 ± 3.19	1.5
5	230.00	227.16 ± 2.60	1.1
6	230.00	225.66 ± 7.80	3.5
7	230.00	236.17 ± 5.20	2.2
8	230.00	222.65 ± 6.88	3.1
9	230.00	234.67 ± 13.51	5.8
10	230.00	212.14 ± 7.80	3.7
Azaperol			
1	230.00	225.06 ± 7.84	3.5
2	230.00	225.06 ± 7.07	3.1
3	230.00	223.10 ± 10.38	4.7
4	230.00	217.22 ± 8.32	3.8
5	230.00	223.10 ± 3.92	1.8
6	230.00	221.80 ± 9.67	4.4
7	230.00	234.21 ± 6.30	2.7
8	230.00	217.22 ± 5.55	2.6
9	230.00	234.21 ± 6.30	2.7
10	230.00	213.30 ± 3.92	1.8

* n = 3.

## Data Availability

HPLC data are available from the authors.

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
