# Peer review of "Development of the Chromatographic Method for Simultaneous Determination of Azaperone and Azaperol in Animal Kidneys and Livers"

_ijms, 2022, doi:10.3390/ijms24010100_

Round 1

Reviewer 1 Report

The article is very well written and organized. However, major revision is required

1-     In abstract , PCA full name should be stated

2-     The following article should be cited and compared with the new method in table form in terms of mobile phase , stationary one, linearity, applications , merits , demerits  

Simultaneous determination of azaperone and azaperol in animal tissues by HPLC with confirmation by electrospray ionization mass spectrometry

https://www.sciencedirect.com/science/article/pii/S1570023208008829?via%3Dihub

https://doi.org/10.1016/j.jchromb.2008.11.047

3-     Novelty statement should be more highlighted in the abstract. Merits over old published methods should be highlighted in the introduction too.

4-     Additionally, the following articles should be cited

http://dx.doi.org/10.1016/j.aca.2006.11.010

https://doi.org/10.1016/j.fochx.2022.100525

5-     Number of references should be increased. I suggest inclusion of review article for analysis of sedatives in animal tissues e.g.

De Brabander, H. F., Herlinde Noppe, Karolien Verheyden, J. Vanden Bussche, Klaas Wille, L. Okerman, Lynn Vanhaecke, Wim Reybroeck, Sigrid Ooghe, and Siska Croubels. "Residue analysis: Future trends from a historical perspective." Journal of Chromatography A 1216, no. 46 (2009): 7964-7976.

Kinsella, Brian, John O’Mahony, Edward Malone, Mary Moloney, Helen Cantwell, Ambrose Furey, and Martin Danaher. "Current trends in sample preparation for growth promoter and veterinary drug residue analysis." Journal of Chromatography A 1216, no. 46 (2009): 7977-8015.

Stolker, A. A. M., and UA Th Brinkman. "Analytical strategies for residue analysis of veterinary drugs and growth-promoting agents in food-producing animals—a review." Journal of chromatography A 1067, no. 1-2 (2005): 15-53.

And many others

6-     gradient profiles in page 5 should be arranged in a table form  or may be added in supplementary file

7-     greenness assessment of the analytical method with comarasion to old ones is strongly recommended using reliable tool e.g. AGREE or GAPI

https://pubs.rsc.org/en/content/articlelanding/2021/ay/d0ay02169e/unauth

 https://pubs.acs.org/doi/10.1021/acs.analchem.0c01887

https://pubs.rsc.org/en/content/articlelanding/2021/GC/D1GC02318G

8-     future perspective should be provided

9-     conclusion should be added 

best wishes

Author Response

Dear Editor, Dear Reviewer,

thank you very much for your valuable comments on our manuscript (ijms-2100473). We appreciate your criticism because it helps to improve our skills in the scientific work.

Following are the changes we have made as well as response to the Reviewer’s comments.

  1. In abstract, PCAfull name should be stated

Ad 1. Corrected.

  1. The following article should be cited and compared with the new method in table form in terms of mobile phase, stationary one, linearity, applications, merits, demerits  

Simultaneous determination of azaperone and azaperol in animal tissues by HPLC with confirmation by electrospray ionization mass spectrometry

https://www.sciencedirect.com/science/article/pii/S1570023208008829?via%3Dihub

https://doi.org/10.1016/j.jchromb.2008.11.047

Ad 2. Thank you for the valuable comment. A comparison with the cited method was added to the manuscript (Table 4).

  1. Novelty statement should be more highlighted in the abstract. Merits over old published methods should be highlighted in the introduction too.

Ad 3. Thank you for the comment. This paper describes the first so fast, simple, and useful method, especially for many laboratories that do not have sophisticated equipment. Taking the above into account, the presented method stands out among other protocols described in the literature primarily by its great usability, but also by its simplicity while maintaining satisfactory validation parameters. This methodology is also not time consuming, which is advantage of described method.

  1. Additionally, the following articles should be cited

http://dx.doi.org/10.1016/j.aca.2006.11.010

https://doi.org/10.1016/j.fochx.2022.100525

Ad 4. The articles mentioned above were cited.

  1. Number of references should be increased. I suggest inclusion of review article for analysis of sedatives in animal tissues e.g.

De Brabander, H. F., Noppe H., Verheyden K., Bussche J.V., Wille K., Okerman L., Vanhaecke L., Reybroeck W., Ooghe S., Croubels S.. "Residue analysis: Future trends from a historical perspective." Journal of Chromatography A 1216, no. 46 (2009): 7964-7976.

Brian K., O’Mahony J., Malone E., Moloney M., Cantwell H., Furey A., Danaher M. "Current trends in sample preparation for growth promoter and veterinary drug residue analysis." Journal of Chromatography A 1216, no. 46 (2009): 7977-8015.

Stolker, A. A. M., and UA Th Brinkman. "Analytical strategies for residue analysis of veterinary drugs and growth-promoting agents in food-producing animals—a review." Journal of chromatography A 1067, no. 1-2 (2005): 15-53.

And many others

Ad 5. Number of references was increased.

  1. Gradient profiles in page 5 should be arranged in a table form  or may be added in supplementary file

Ad 6. Corrected.

  1. Greenness assessment of the analytical method with comarasion to old ones is strongly recommended using reliable tool e.g. AGREE or GAPI

https://pubs.rsc.org/en/content/articlelanding/2021/ay/d0ay02169e/unauth

 https://pubs.acs.org/doi/10.1021/acs.analchem.0c01887

https://pubs.rsc.org/en/content/articlelanding/2021/GC/D1GC02318G

Ad 7. Green analytical chemistry focuses on making analytical procedures more environmentally benign and safer to humans [1]. We decided to check how the proposed method looks like in this matter. We used very smart software dedicated for this purpose, i.e. the Analytical GREEnness calculator for the assessment of greenness of analytical procedures based on the SIGNIFICANCE principles [1]. The calculated greenness of presented method is 0.58. Our procedure is based on combined sample homogenization and analytes extraction, and separation of analytes by HPLC with UV detection. The procedure consisted of external sample treatment with reduced number of steps (principle 1), and 0.5 g of tissue sample is needed (principle 2). The measurement is off-line (principle 3), and the procedure involves three distinct steps such as homogenization, lyophilization, centrifugation and separation (principle 4). The procedure is semi-automated and miniaturized (principle 5). During the analysis no derivatization step was required (principle 6). Analytical wastes include 2 mL of acetonitrile for homogenization, 20 µL of ethanol for dilution, 15 µL of PCA for deproteinization as well as 10.1 mL (including 2.9 mL of acetonitrile) of the HPLC mobile phase (principle 7). Two analytes are determined in a single run and the sample throughput is ∼4 samples per hour, if we assume that about 40 samples can be prepared simultaneously (principle 8). Lyophilization system is the most energy-demanding analytical technique in our protocol (principle 9). Some of the reagents can be from bio-based sources (principle 10). The procedure requires 4.9 mL of toxic solvents (principle 11), and acetonitrile is considered explosive (fumes) and highly flammable (principle 12). To the manuscript greenesses comparison with other methods was added.

[1] Pena-Pereira, F., Wojnowski, W., Tobiszewski, M., AGREE - Analytical GREEnness Metric Approach and Software, Anal. Chem. 2020, 92, 10076−10082. https://dx.doi.org/10.1021/acs.analchem.0c01887

  1. Future perspective should be provided

Ad 8. Due to the very good precision and accuracy of the method, we strongly believe that it can be used in the future for the routine analysis of meat for the content of azaparone very often administered to animals, and its metabolite azaperol. The described method can be used both in veterinary medicine and in food safety testing. Since, eating residues of these compounds with meat when the withdrawal period has not been respected is hazardous to the health of consumers the method may be helpful in protecting human health.

  1. Conclusion should be added 

Ad 9. Thank you for the valuable comment. Conclusions have been added. Unfortunately, they did not add when we uploaded the manuscript.

Reviewer 2 Report

The present manuscript entitled "Development of the chromatographic method for simultaneous determination of azaperone and azaperol in animal kidneys and livers" by Izabella Kośka and Paweł Kubalczyk (ijms-2100473) is written correctly and has a good structure; moreover, it has all the necessary parts. The article is interesting from an analytical and medical point of view; therefore, it should interest the reader. I proposed improvements in the method description and with a presentation of figures. The paper meets the International Journal of Molecular Sciences' requirements, and I recommend the article for publication in the International Journal of Molecular Sciences following the common editing stage. My current decision is a major revision. More specific comments and observations are presented below.

1. Please check that all abbreviations are derived before using them.

2. Figure 1 and Figure 2 can be combined into one drawing.

3. Page 2, line 72. Figure S3 is mentioned, but Figure 3 should be.

4. The axes should be corrected in the drawings to make them more visible.

5. Please check the record of units and unify them (“l” or “L”).

6. The “relationship” is mentioned. This term should be changed to "relation". The relationship tends to be used more broadly to describe the interactions between specific people or smaller groups of people.

7. Page 4, lines 98-111. This fragment should be assigned to a separate section along with the title.

8. Section 2.1.3. should be expanded.

9. Has the interference been studied? What can be done in the event of strong interference effects? How would you deal with them? What types of interference effects could occur?

10. RSD expressed as a percentage is the coefficient of variation (CV).

11. Page 6, line 187. Please give specific concentrations, not just a range.

12. Appropriate tools should be used to best characterize the method when developing a new approach (e.g., AGREE- Analytical GREEnness Metric Approach or RGB model).

13. The results presented in Table 3 should be discussed in detail in the text.

14. Section 3.1. What were the parameters of the water used?

15. Does the developed method have disadvantages?

16. Please correct the typos in the text.

17. Please add a Conclusions section.

I hope that the comments presented will help improve the article.

Author Response

Dear Editor, Dear Reviewer,

thank you very much for your valuable comments on our manuscript (ijms-2100473). We appreciate your criticism because it helps to improve our skills in the scientific work.

Following are the changes we have made as well as response to the Reviewer’s comments.

  1. Please check that all abbreviations are derived before using them.

Ad 1. Checked and corrected.

  1. Figure 1 and Figure 2 can be combined into one drawing.

Ad 2. Corrected.

  1. Page 2, line 72. Figure S3 is mentioned, but Figure 3 should be.

Ad 3. Corrected.

  1. The axes should be corrected in the drawings to make them more visible.

Ad 4. Corrected.

  1. Please check the record of units and unify them (“l” or “L”).

Ad 5. Checked and corrected.

  1. The “relationship” is mentioned. This term should be changed to "relation". The relationship tends to be used more broadly to describe the interactions between specific people or smaller groups of people.

Ad 6. Thank you for the comment. Corrected.

  1. Page 4, lines 98-111. This fragment should be assigned to a separate section along with the title.

Ad 7. Corrected.

  1. Section 2.1.3. should be expanded.

Ad 8. PCA, trichloroacetic acid, or excess acetonitrile can be used to deproteinize the samples. Since PCA is one of the most popular deproteinization agent, we decided to use it in our methodology. This is a very important step in sample preparation, because proteins present in a sample could easily block the chromatography column. At the same time, it is essential not to add unnecessary excess of deproteinizing reagent, as it further dilutes the sample resulting in poorer LOQ. For this purpose, we checked the volume of PCA at which complete deproteinization occurs. The following volumes of 3 mol/L PCA were tested: 5, 10, 15, 20, 25 and 30 µL. After the addition of 15 µl of PCA, we noticed that adding another volume of PCA did not cause further proteins precipitation. Therefore, such a volume of acid was selected for the experiments to avoid excessive dilution of the sample and ensure its complete deproteinization.

  1. Has the interference been studied? What can be done in the event of strong interference effects? How would you deal with them? What types of interference effects could occur?

Ad 9. Thank you for the valuable comment. In developing the method, we used a large number of different animal tissue samples that had been purchased over a long period of time and we did not notice any interference. Azaperone is the sedative drug most commonly used in animals for this purpose. Unfortunately, we do not have standards for other drugs in this group to check their potential interference. However, it is very unlikely that an animal will receive several sedatives at the same time. In addition to sedatives, farm animals may also receive other medications, such as antibiotics, such as fluoroquinolones, to treat infections or prevent them. Several of these drugs (i.e. ofloxacin and ciprofloxacin) were checked by us for potential chromatogram interference and we did not notice any. However, if the tissue samples contain a large number of interfering agents, a modification of the extraction step could be necessary during sample preparation.

  1. RSD expressed as a percentage is the coefficient of variation (CV).

Ad 10. Corrected.

  1. Page 6, line 187. Please give specific concentrations, not just a range.

Ad 11. Corrected.

  1. Appropriate tools should be used to best characterize the method when developing a new approach (e.g., AGREE- Analytical GREEnness Metric Approach or RGB model).

Ad 12. Green analytical chemistry focuses on making analytical procedures more environmentally benign and safer to humans [1]. We decided to check how the proposed method looks like in this matter. We used very smart software dedicated for this purpose, i.e. the Analytical GREEnness calculator for the assessment of greenness of analytical procedures based on the SIGNIFICANCE principles [1]. The calculated greenness of presented method is 0.58. Our procedure is based on combined sample homogenization and analytes extraction, and separation of analytes by HPLC with UV detection. The procedure consisted of external sample treatment with reduced number of steps (principle 1), and 0.5 g of tissue sample is needed (principle 2). The measurement is off-line (principle 3), and the procedure involves three distinct steps such as homogenization, lyophilization, centrifugation and separation (principle 4). The procedure is semi-automated and miniaturized (principle 5). During the analysis no derivatization step was required (principle 6). Analytical wastes include 2 mL of acetonitrile for homogenization, 20 µL of ethanol for dilution, 15 µL of PCA for deproteinization as well as 10.1 mL (including 2.9 mL of acetonitrile) of the HPLC mobile phase (principle 7). Two analytes are determined in a single run and the sample throughput is ∼4 samples per hour, if we assume that about 40 samples can be prepared simultaneously (principle 8). Lyophilization system is the most energy-demanding analytical technique in our protocol (principle 9). Some of the reagents can be from bio-based sources (principle 10). The procedure requires 4.9 mL of toxic solvents (principle 11), and acetonitrile is considered explosive (fumes) and highly flammable (principle 12).

[1] Pena-Pereira, F., Wojnowski, W., Tobiszewski, M., AGREE - Analytical GREEnness Metric Approach and Software, Anal. Chem. 2020, 92, 10076−10082. https://dx.doi.org/10.1021/acs.analchem.0c01887

  1. The results presented in Table 3 should be discussed in detail in the text.

Ad 13. The results obtained and collected in the Table 3 indicate that the described methodology can be successfully used for routine analysis of meat for the content of azaperone and azaperol. The obtained results are repeatable and consistent.

  1. Section 3.1. What were the parameters of the water used?

Ad 14. Deionized water (Type 1) was obtained using a Millipore Milli-Q-RG System (Waterford, Ireland) with a resistivity of 18 kW•cm at 25 °C. The water was filtered using a membrane filter with a pore diameter of 0.22 mm.

  1. Does the developed method have disadvantages?

Ad 15. The developed method has disadvantages, too. The weaknesses of the presented method include, above all, the use of large volumes of the toxic solvent acetonitrile, as well as small concentration sensitivity when using UV-Vis detection what required the need to concentrate the sample.

  1. Please correct the typos in the text.

Ad 16. Checked and corrected.

  1. Please add a Conclusions section.

Ad 17. Conclusions have been added, unfortunately they did not add when we uploaded the manuscript.

Round 2

Reviewer 1 Report

thanks for your professional work  and productive efforts. the paper could be published in the current form .

best wishes 

Reviewer 2 Report

Dear Authors,

Thank you for your meticulous consideration of my comments. The paper has improved substantially and, in my opinion, is suitable for publication.